# Eco-Epidemiological Evidence of the Transmission of Avian and Human Influenza A Viruses in Wild Pigs in Campeche, Mexico

**DOI:** 10.3390/v12050528

**Published:** 2020-05-11

**Authors:** Brenda Aline Maya-Badillo, Rafael Ojeda-Flores, Andrea Chaves, Saul Reveles-Félix, Guillermo Orta-Pineda, María José Martínez-Mercado, Manuel Saavedra-Montañez, René Segura-Velázquez, Mauro Sanvicente, José Iván Sánchez-Betancourt

**Affiliations:** 1Laboratorio de Investigación del Departamento de Medicina y Zootecnia de Cerdos, Facultad de Medicina Veterinaria y Zootecnia (FMVZ), Universidad Nacional Autónoma de Mexico, Ciudad Universitaria, Av. Universidad #3000, Ciudad de Mexico 04510, Mexico; mayis.bamb20@gmail.com (B.A.M.-B.); mvzsaulreveles@hotmail.com (S.R.-F.); 2Laboratorio de Ecología de Enfermedades y Una Salud, Departamento de Etología, Fauna Silvestre y Animales de Laboratorio, FMVZ, UNAM, Ciudad de México 04510, Mexico; ojedar@unam.mx (R.O.-F.); ortapg@outlook.com (G.O.-P.); 3Department of Medicine and Epidemiology, School of Veterinary Medicine, University of California, Davis, CA 95616, USA; andreachaves.biol@gmail.com; 4Unidad de Investigación, FMVZ, UNAM, Ciudad de México 04510, Mexico; majosemm13@gmail.com (M.J.M.-M.); realselab@yahoo.com.mx (R.S.-V.); 5Departamento de Microbiología e Inmunología, FMVZ, UNAM, Ciudad de México 04510, Mexico; manuelsaavedra76@gmail.com; 6Colegio de Postgraduados, Campus Puebla, Carretera federal Mexico-Puebla km. 125.5, (Boulevard Forjadores), Puebla C.P. 72760, Mexico; sanvicentemauro@yahoo.com.mx

**Keywords:** A/H1N1, A/H5N2, synanthropic, wild pig, zoonosis

## Abstract

Influenza, a zoonosis caused by various influenza A virus subtypes, affects a wide range of species, including humans. Pig cells express both sialyl-α-2,3-Gal and sialyl-α-2,6-Gal receptors, which make them susceptible to infection by avian and human viruses, respectively. To date, it is not known whether wild pigs in Mexico are affected by influenza virus subtypes, nor whether this would make them a potential risk of influenza transmission to humans. In this work, 61 hogs from two municipalities in Campeche, Mexico, were sampled. Hemagglutination inhibition assays were performed in 61 serum samples, and positive results were found for human H1N1 (11.47%), swine H1N1 (8.19%), and avian H5N2 (1.63%) virus variants. qRT-PCR assays were performed on the nasal swab, tracheal, and lung samples, and 19.67% of all hogs were positive to these assays. An avian H5N2 virus, first reported in 1994, was identified by sequencing. Our results demonstrate that wild pigs are participating in the exposure, transmission, maintenance, and possible diversification of influenza viruses in fragmented habitats, highlighting the synanthropic behavior of this species, which has been poorly studied in Mexico.

## 1. Introduction

The ecological and evolutionary traits of influenza A virus make it unique in many respects, from the great diversity of hosts it infects (wild birds, poultry, swine, equines, canines, sea mammals, and humans, among others) to its distinctive capacity to evolve and adapt itself, making interspecies leaps and establishing in new populations in some cases. Both traits are due to a segmented genome, which gives the virus a remarkable evolutionary capacity; this capacity is much higher in viral surface glycoproteins, but it is evident (to a lesser degree) in all eight viral genome segments [1,2,3]. Considering such multi-host complexity, most studies have focused on domestic animals and humans, due to the importance of the infection in animal and public health. In contrast, few works have been conducted on influenza A virus in wild hosts. To date, 16 hemagglutinin (HA) and 9 neuraminidase (NA) variants have been recognized in wild bird viruses [4,5], and the isolation of H17N10 and H18N11 virus subtypes has been reported in rectal swabs from the bat species *Sturnira lilium* and *Artibeus jamaicensis* [2,6]. Despite these findings, our understanding of the interaction between influenza A virus subtypes and wild hosts is still limited. Most studies on infectious agents like influenza A viruses in wild pigs (*Sus scrofa*) have been conducted in the USA. Only one study on this host has been published in Mexico, covering the northern region of the country [7].

Wild pigs [8] originated from several human-linked activities. Wild pigs are domestic pigs that are released or escape captivity, European boars, and hybrids of these two when they coincided in time and space [9,10]. In the case of domesticated pig populations that escaped into the wild and established in various habitats, phenotypic traits of the original agriotype, the Eurasian boar, have reappeared over time and generations. These populations have interbred with the European boar in places where the latter was translocated and regarded as an exotic species; thus, the phenotype of these animals is extremely diverse [9,10,11,12,13,14,15,16]. Wild pigs live in all continents except for Antarctica, and stable populations are currently found throughout the world [7,11,16,17,18]. This wide distribution is due to the anthropogenic introduction of domesticated pigs and Eurasian boars as a food source or for sport hunting, respectively [10]. Wild pigs are synanthropic, a feature exhibited by non-domesticated animals to interact in the human and wild environments at the same time, have a great invasive capacity in new areas [19], and cause severe damage to the environment [12]. Despite its severe repercussions at a socioeconomic, ecological, and epidemiological level, on public and animal health, and even its impact on protected natural areas—as it is the case of Área de Protección de Flora y Fauna Laguna de Términos (APFFLT), habitat of various endangered species—the presence of influenza A virus subtypes in wild pigs in Central and Southern Mexico has not been studied to date.

Wild pigs are known to be a generalist species that interacts with wild and domestic animals and with humans. Since it is a susceptible host to infection by influenza A virus, this study is aimed to determine whether the population of wild pigs in the Laguna de Términos region and nearby areas is participating in the transmission and maintenance of influenza A virus subtypes.

## 2. Material and Methods

### 2.1. Area Under Study

All samples were collected in Carmen and Palizada, municipalities in southwest Campeche. Site selection was based on the presence of wild pigs, either recently captured or recently hunted animals. Parts of the APFFLT were also included in the area under study, with the collaboration and authorization of the National Commission of Protected Natural Areas (CONANP), since the presence of wild pigs had been reported in these areas. This region is part of a great fragment of natural vegetation, formed by a flood plain with an altitude between 0 and 3 m above mean sea level, mainly covered by flood plant associations of palms, tropical deciduous, and sub-deciduous forest, along with natural flood savannas and marshes. Mangrove associations and flood plants like arrowroot (*Thalia geniculata, Pontederia sagittata* and *Sagittaria lancifolia*) and cattail *(Thypha dominguensis*) grow on the banks of the rivers. In total, 20 sites were selected for sampling, 17 of which were in the Carmen municipality and 3 were in the Palizada municipality, located inside the APFFLT (Figure 1).

### 2.2. Ethical Considerations

The collection and handling of biological samples from wild pigs in this study were approved by the Institutional Subcommittee for the Use and Care of Experimental Animals (SICUAE), permit No. MC-2018/2-11.

### 2.3. Sample Categorization

The sampling was carried out from May to October 2019. It began in the dry season and ended in the rainy season. Sampled animals were divided into two groups, caught and captive; wild pigs that had to be caught for sampling, most of which were regarded as minor game, were included in Group A. Captive wild swine that had roamed free but were previously caught and had spent from one day to 4 weeks in confinement were included in Group B (Appendix A). Wild pigs were also classified according to phenotype. Morphometric determinations were performed, including thorax circumference (TC), total length (TL), height at the withers (HW), snout length (SL), and ear length (EL). Hogs with wild phenotype, i.e., medium-large size with respect to domestic swine; barrel-shaped, sturdy thorax (often flattened on the sides); short, thin hind/fore limbs; a relatively long body; a pointed head at the end of a short neck; black, smooth coat with bristly hair across the back; small eyes; relatively large and wide, hairy ears narrowing toward the tip; a short tail, either straight or curly and covered with hair, especially at the tip [10], were included in Group 1. Wild pigs with a mixed phenotype, i.e., meeting some but not all traits of a wild phenotype and showing some physical traits of a domesticated pig, were included in Group 2 (Appendix A). Discrete categorical variables like sex (female of male), age (piglet, juvenile, or adult) and reproductive status (reproductive or non-reproductive) were also recorded for all animals.

### 2.4. Sample Collection, Transport, and Storage

Nasal swabs were collected from wild pigs in Groups A and B by inserting a swab into the nasal cavity of each animal and rotating it against the mucosal walls; the swabs were then placed in a 15-mL plastic tube containing phosphate buffered saline solution (PBS), frozen in liquid nitrogen (−195.8 °C) and stored at −70 °C until processed. Additionally, key organs involved in infection by influenza A virus, like trachea, lung, and mediastinal lymph nodes, were collected, placed in plastic beakers, frozen in liquid nitrogen (−195.8 °C), and stored at −70 °C until processed. Five-milliliter blood samples were collected directly from the jugular vein of hogs in both groups in anticoagulant-free blood-collection tubes and kept at 4 °C. Then, serum was obtained by centrifugation and stored at −20 °C until processed.

### 2.5. Hemagglutination Inhibition

The standard procedure for the hemagglutination inhibition (HI) assay recommended by the World Organisation for Animal Health (OIE) [20] was followed in this study, except that 8 hemagglutinating units (HAU) were used. Serum samples were heat-inactivated at 56 °C and pretreated with kaolin and 0.5% chicken (*Gallus gallus*) erythrocytes. Serial double dilutions were performed, and serum titers were regarded as positive when titers were equal to or higher than 1:80 HIU. For this assay, the following influenza virus subtypes were used as antigen source: Human H1N1 (ID A/Mexico/INER1/2000 (H1N1), GenBank accession No. JN086908), swine H1N1 (ID A/swine/Mexico/JalDMZC05/2015 (H1N1), GenBank accession No MH013200-MH013207), swine H1N2 (ID A/swine/MexMich/DMZC/2014 (H1N2)), swine H3N2 (ID A/swine/Mexico/HgoDMZC11/2015 (H3N2), GenBank accession No. MH006715-MH006722), and avian H5N2 (ID A/swine/Mexico/EdoMexDMZC03/2015 (H5N2), GenBank accession No. MH013208-MH013215).

### 2.6. Extraction and Molecular Detection of Viral RNA

Nasal swab samples were centrifuged at 5000 rpm for 10 min. Then, supernatants were collected and used for RNA extraction with QIAamp cador Pathogen Mini Kit (Qiagen, Hilden, Germany), following the manufacturer’s instructions. Organ samples were macerated with liquid nitrogen and a 1:10 suspension in PBS was prepared. These suspensions were vigorously shaken for 5 min and centrifuged at 5000 rpm for 15 min. The supernatants were used for RNA extraction as described above. Viral RNA was detected by qRT-PCR, using VetMAX-Gold SIV Detection Kit (Life Technologies, Carlsbad, CA, USA) targeting the matrix (*M*) and nucleoprotein (*NP*) genes.

### 2.7. Virus Isolation

For viral replication, the supernatants of positive samples were cultured in specific pathogen-free (SPF); 9–11 days-old chicken embryos (CE). Each embryo was inoculated via the allantoic cavity with 200 μL of supernatant, and the allantoic fluid was collected at 48 and 72 h post-inoculation [21].

### 2.8. Sequencing

An RT-PCR assay to amplify the eight segments of the viral genome was performed on those nasal swab and organ samples that were positive for qRT-PCR and viral isolation, using SuperScript III One-Step RT-PCR System with Platinum Taq High Fidelity DNA Polymerase kit (Thermo Fisher Scientific, Waltham, MA, USA) and an end-point thermocycler. RT-PCR products were resolved by 2% agarose gel electrophoresis. Then, libraries were prepared using the Ion Xpress Plus Fragment Library Kit (Thermo Fisher Scientific), following the manufacturer’s instructions under a 200-base pair (bp) protocol. Library molarity was determined with the High Sensitivity DNA Kit (Agilent, Santa Clara, CA, USA). An equimolar library mixture was prepared, and the template was constructed and subjected to clonal amplification by emulsion PCR, using Ion PGM Hi-Q View OT2 Kit Ion Torrent kit (Thermo Fisher Scientific) on an Ion OneTouch 2 System. Then, templates were sequenced with the Ion PGM Hi-Q View Sequencing Kit (Thermo Fisher Scientific) on the Ion Torrent Genome Machine (PGM) platform. Readings with a Q Score > 20 were selected for sequence assembly.

### 2.9. Phylogenetic Analysis

The sequences of influenza A virus subtypes were compared with those reported in GenBank, using NCBI BLAST and selecting those sequences with an identity > 95%. The phylogenetic analysis was performed independently for each genome segment with the software MEGA v.10.1, using the maximum verisimilitude method, to infer in the evolutionary history. Finally, the phylogenetic trees were edited with the software FIGTREE v.1.4.3 [22].

### 2.10. Categorizing the Degree of Landscape Anthropization

The degree of anthropization was determined by characterizing the landscape vegetation cover with the software ArcGIS v.10.5. Following the work by Martínez (2010), a score based on the integrated relative anthropization index (IRAI) was used, considering the vegetation cover in the Laguna de Términos region and nearby areas [23]. To estimate the anthropization degree, SENTINEL-2A satellite images were used, in the panchromatic band No. 8, with a resolution of 10 m per pixel. Land use was divided into four classes (Forest, Agricultural Land, Urban Area, and Water Body) according to the spectral signature of the vegetation cover. Five polygons were drawn for each class for a finer selection. Then, eight analysis units (AU), i.e., the areas for which anthropization index was assessed, were defined. These areas were selected according to the distribution of wild pigs, establishing a mean buffer area of 25.51 km^2^ between male and female hogs [9,10,24,25,26]. Then, each AU was subdivided into equal parts, named as analysis subunits (ASU); in this work, each ASU is represented by one pixel [23]. An anthropization index was assigned to each ASU based on the classification described above, obtaining three main categories. The category “Forest” included tropical forest, marshes, and water flows; the category “Agricultural land” included crop fields, pastures, and grazing lands; the category “Urban area” included human settlements, areas with no vegetation cover, highways, and roads [23,27]. Anthropization indices are inherent to the area under study, reflecting the relative nature of IRAI (Appendix A). Once anthropization indices were assigned to individual ASUs, the relative degree of anthropization was estimated for each AU by Equation (1):
(1)$$\mathrm{IRAI}=\frac{\sum^{\;}\mathrm{ASU}}{n}\times 100$$
where ΣASU is the sum of the anthropization index values for all ASUs, and *n* is the total number of ASUs in the AU.

### 2.11. Statistical Analysis

HI seroprevalence was defined as the proportion of wild swine that were positive to the HI assay with respect to the total number of animals assayed, estimated with a confidence level of 95%. Similarly, molecular detection seroprevalence was defined as the proportion of wild pigs that were positive to the qRT-PCR assay with respect to the total number of animals assayed, estimated with a confidence level of 95%. A logistic regression model was also estimated, using the statistical software R, with α = 0.05. The response variable was binary: Positive or negative individuals to HI or molecular detection assays; the explanatory (independent) variables were continuous, quantitative variables like IRAI, home range, and morphometric values, as well as discrete, qualitative variables like sex, age, and phenotype [28].

The logistic regression model is based on the logit function, defined as the natural logarithm (ln) of the probability of failure:(2)$$\mathrm{logit}\left(p\right)=\ln\frac{p}{1-p}$$
where *p* is the probability of failure.

When more than one explanatory (quantitative or qualitative) variable is used, the model is expressed as follows:(3)$$\mathrm{logit}\left(p\right)=a+b_{1}x_{1}+b_{2}x_{2}+\ldots +b_{i}x_{i}$$
where *p* is the probability of failure, and *x*_1_, *x*_2_, … *x_i_* are explanatory variables.

A chi-squared test for variable independence was also performed for discrete, qualitative variables in our study (sex, age, and phenotype) with respect to HI and molecular detection results, with α = 0.05.

The function used for the chi-squared model was
(4)$${Χ}^{2}=\sum_{i-1}^{n}\frac{{\left(O_{i}-E_{i}\right)}^{2}}{E_{i}}$$
where *O* is the observed frequency, and *E* is the expected frequency, with the sum taken over all *i* observations.

Similarly, a *t*-test was performed to compare mean morphometric values between Group 1 and Group 2, considering the continuous, quantitative variables TC, TL, HW, EL, and SL, with α = 0.05.

The function used for the Student’s *t*-model was
(5)$$t_{n+m-2}=\frac{{\bar{x}}_{1}-{\bar{x}}_{2}}{\mathrm{SEDM}}$$

Where SEDM is the standard error of the difference between two means.

## 3. Results

### 3.1. Categorizing Sampled Animals

In total, 61 animals were sampled. From them, 35 (57.37%) were included in Group A (free roaming) and 26 (42.63%) were included in Group B (in confinement). From the total population, 25 animals (40.98%) were included in Group 1 (wild phenotype) and 36 (59.02%) were included in Group 2 (mixed phenotype). No statistically significant differences were found between Groups 1 and 2 with respect to phenotype-associated morphometric values. From the total of sampled animals, 33 (54.1%) were females and 28 (45.9%) were males; with respect to age, 23 (37.7%) were piglets, 18 (29.5%) juveniles, and 20 (32.78%) adults. Finally, 38 animals (62.3%) were non-reproductive and 23 (37.7%) were reproductive.

### 3.2. Hemagglutination Inhibition

From the 61 sampled individuals, 9 were positive to one or more influenza A subtypes with a titer higher than 80 HIU, the chosen cutoff value. Seven animals showed seroconversion to human H1N1 virus, 5 to swine H1N1, and one to avian H5N2, with titers ranging from 80 to 2560 HIU, as shown in Table 1. From the 9 positive individuals, 4 were females and 5 were males. With respect to age, 4 were piglets, 2 were juveniles, and 3 were adults. Four animals showed simultaneous seroconversion for two viral subtypes, human H1N1 and swine H1N1, while 3 wild pigs only showed seroconversion to the human H1N1 viral subtype, one animal only showed seroconversion to the swine H1N1 subtype and one only to the avian H5N2 subtype. Total seropositivity was estimated to be 14.75% (CI 5.85%–23.65%). Seroprevalence for the human H1N1 subtype was 11.47%; for the swine H1N1 subtype it was 8.19%; and for the avian H5N2 subtype it was 1.63%. No positive seroconversion was observed for the swine H1N2 nor for the swine H3N2 viral subtype (Table 1). No other animal showed seroconversion to any viral subtype, with titers less than the cutoff value of 80 HIU.

### 3.3. qRT-PCR of Influenza A Virus M Gene

In total, 12 samples from 12 different animals were positive to the qRT-PCR assay. From them, 8 were nasal swabs, 3 were tracheal samples, and one was a lung sample, as shown in Table 2. The samples with the highest viral RNA load were one nasal swab, with a Ct value of 28.43, and one tracheal sample, with a Ct value of 31.20. From the individuals that were positive to this assay, 6 were females and 6 were males; with respect to age, 3 were piglets, 2 were juveniles, and 7 were adults. Total prevalence by molecular detection was estimated to be 19.67% (CI 9.70%–29.65%) (Table 2).

### 3.4. Virus Isolation

From all 12 positive samples, only one tracheal sample from an adult, male wild pig with a titer of 1:16 HAU and one tracheal sample from another adult, male wild pig with a titer of 1:32 HAU were positive for virus isolation. The virus titer in the first sample increased to 1:64 HAU on the second passage, while the titer of the second sample decreased to 8 HAU on the second passage. On the third passage, the titer of the first sample remained unaltered, while the titer of the second sample decreased to 4 HAU. None of the other 10 samples that were positive to qRT-PCR replicated upon inoculation to chicken embryo, and no positive titer was observed on the first passage.

### 3.5. Sequencing and Phylogenetic Analysis

Complete genome sequences were obtained from one tracheal isolate and from a direct tracheal sample from the same animal. No other sample positive to qRT-PCR amplified for the eight genome segments of influenza virus. Libraries for sequencing were not obtained for any of those samples nor for the second viral isolate.

Sequencing analysis and inferred topologies for the *HA* gene showed that the HA protein from the Feral swine/Campeche/DMZC-DEFSAL-UIFMVZ19-12 (H5N2) virus is in the same clade as swine H5N2 influenza virus subtypes that had been reported in Veracruz and State of Mexico in 2014 and 2015, respectively, and as avian H5N2 influenza viruses isolated from poultry (particularly chickens) in Mexico in 1994 and 1995. The tree shows two main clades, one including avian H5N2, avian H5N4, avian H5N9, and avian H5N1 virus subtypes, primarily isolated in North America from wild birds like mallards (*Anas platyrhynchos*) and tundra swans (*Cygnus columbianus*) but also found in poultry like turkeys (*Meleagris gallopavo*) and chickens (*G. gallus*). The other main clade is entirely composed by avian and swine H5N2 virus subtypes mostly isolated in Mexico, although some of them originated in Taiwan. This clade includes viruses isolated from poultry, specifically chickens, and from pigs (*S. scrofa*). The HA sequence isolated from a wild pig and reported herein is included in this second clade (Figure 2).

Sequencing analysis and inferred topologies for the *NA* gene showed that the NA protein from the Feral swine/Campeche/DMZC-DEFSAL-UIFMVZ19-12 (H5N2) virus is in the same clade as Mexican swine H5N2 virus subtypes isolated in Guanajuato in 2014 and State of Mexico in 2015, and as avian H5N2 influenza viruses isolated in Mexico in 1994 and 1995. The tree shows two main clades, one including avian H5N2 and avian H6N2 virus subtypes mostly isolated in Mexico, although some of them originated in Guatemala and a few in the USA in poultry like chickens (*G. gallus*) and turkeys (*M. gallopavo*). The other main clade is entirely composed by avian and swine H5N2 virus subtypes mostly isolated in Mexico, although some of them originated in Taiwan. This clade includes viruses isolated from poultry, specifically chickens, and pigs (*S. scrofa*). The NA sequence isolated from a wild pig and reported herein is included in this second clade (Figure 3).

Sequencing analysis and inferred topologies for the *PB2, PB1, PA, NP, M,* and *NS* genes showed that the proteins from the Feral swine/Campeche/DMZC-DEFSAL-UIFMVZ19-12 (H5N2) virus are in the same clade as swine H5N2 influenza virus subtypes isolated in Mexico in 1994 and 1995, with particular differences in each protein, as shown in the phylogenetic trees for each genome segment; these differences had been reported in virus subtypes isolated in pigs from intensive production farms in Guanajuato in 2014 and State of Mexico in 2015 (Appendix A).

### 3.6. Categorizing the Degree of Landscape Anthropization

In general, the dominant vegetation cover category, with 50% of global frequency, was “Forest”, the least anthropized category; it includes tropical forest, marshes, aquatic vegetation, swamps, and water bodies. However, the category “Agricultural land” is very close in proportion to “Forest”, with 45%. Figure 4 shows the proportion of vegetation cover categories in each AU. As shown, the most frequent vegetation cover category by AU or quadrant is “Agricultural land”; this indicates that the habitat is highly fragmentated, and forested areas are adjacent to crop and livestock production areas. From the eight AUs, the least anthropization degree (lowest IRAI values) was observed in C1, which includes well-conserved areas within APFFLT, with marshes, aquatic vegetation, and fragments of tropical forest (high evergreen forest, medium sub-deciduous forest, and low evergreen forest) as the predominant vegetation cover; however, there are patches of grazing land used for livestock production in the natural protected area. The AU with highest IRAI was C6 due to the presence of areas producing crops like oil palm (*Elaeis guineensis*), sorghum (*Sorghum* sp.), and corn (*Zea mays*), as well as grazing areas, human settlements and roads.

### 3.7. Statistical Analysis

#### 3.7.1. Logistic Regression Model

A logistic model using seropositivity as the response variable and IRAI as the explanatory variable showed an effect of landscape anthropization on seropositivity, with a marginal significance, *p* = 0.0576 (Figure 5). No other variable showed a similar trend.

#### 3.7.2. Chi-Squared (χ^2^)

Our results show that the variables age and qRT-PCR positivity are dependent, with *p* = 0.05. No statistically significant association was observed among other variables.

## 4. Discussion

The eco-epidemiology of emerging and re-emerging, multi-host zoonotic diseases like influenza A has been little addressed in epidemiologic surveillance programs in Mexico. Few works have studied the possibility of interspecies transmission, and most reports have focused on detecting the infective agent in one host at a time, particularly in domestic hosts. Even fewer studies have addressed the link of anthropogenic changes like landscape anthropization and, to a lesser degree, the introduction of exotic species with the ecology of a virus of importance in public and animal health, like the influenza A virus. In this study, we demonstrate the exposition of wild pigs to influenza A viruses of swine, avian, and mainly human origin, with antibody titers ranging from 80 to 2560 HIU for human H1N1, swine H1N1, and avian H5N2 subtypes.

In a study conducted from 2010 to 2013, serum samples were obtained from wild pigs in 35 states in the USA and analyzed against 45 circulating influenza A virus antigenic variants of human, swine, and avian origin. From the population sampled, 4.9% were seropositive to influenza A virus [29]. In our study, 14.7% of sampled hogs were positive. Some animals showed antibodies against both human and avian viruses; this is noteworthy, since it proves that wild pigs are susceptible to infection by virus subtypes originated in other species, even living in the wild; furthermore, this could result in the emergence of new influenza virus subtypes. A seroprevalence rate of 30.7% was observed in wild pigs from the Biosphere Reserve Sierra La Laguna, in Baja California Sur, Mexico; the viral variants identified included the swine H1N1 and H3N2 subtypes used in this study [7]. The differences in the seroprevalence values reported in other works and in this study indicate that other factors could favor a higher or lower contact of wild pigs with different influenza A virus subtypes, including the time of the year, environmental characteristics, and especially the interaction with other hosts susceptible to infection by influenza A virus, like wild birds and humans [30].

Martin et al. (2017) also found double-positive individuals for different swine and avian virus subtypes; this indicates that wild pigs were exposed to both swine and avian influenza A viruses. In our study, we found, additionally, wild pigs that were double positive for human and swine H1N1 virus subtypes. This indicates that wild pigs could be exposed to infection by influenza virus variants of different origins, demonstrating the risk of viral reassortment in wild pigs and the emergence of novel variants with a negative impact on animal or public health [2,29,31].

A major concern is the exposition of wild pigs to human viruses. While the 2009 pandemic evinced the importance of the interaction between human and swine hosts, the fact that the transmission of influenza virus from humans to pigs is much more frequent than its converse and the relevance of the former in the viral diversity observed in swine hosts have not been emphasized enough [32]. Globally, the transmission of human influenza A virus subtypes to pigs is the main inverse zoonosis—Anthropozoonosis—Of an infective agent documented in other studies. The bias against pigs, regarding them as infectious sources for humans instead of susceptible hosts, is key to understand the bidirectional nature of the human–animal interface and how this interaction leads to infection risks for all hosts involved [33,34]. Our finding of titers up to 2560 HIU for human H1N1 virus subtype in wild pigs evinces the exposure and infection of these animals by influenza viruses of human origin. This interaction between wild pigs and humans in the Laguna de Términos region exhibits the zoonosis/anthropozoonosis taking place in this system and shows that wild pigs are not merely influenza A virus transmitters, but they can also be exposed, infected, and get ill, at least for different virus subtypes of human origin.

The exposure to influenza A viruses and their transmission to humans is favored by hunting and capture of wild pigs, or through dogs used for hunting [35]. Human activities are promoting synanthropy in wild pigs. Additionally, there is growing evidence of human translocation of wild pigs [36] for sports hunting or backyard rearing; unfortunately, the biosafety measures applied are not appropriate, and the physical and evolutionary traits of these animals often allow them to escape, re-establishing in new areas and interacting again with wild hosts, with the added risk of having been exposed to infective agents from humans and domestic animals, like influenza viruses [37,38,39]. Wild pigs in our study showed exposure to swine H1N1 virus with antibody titers from 80 to 320 HIU and to the avian H5N2 subtype with a titer of 80 HIU. While these titers are low compared to the human subtype, they prove the interaction with these virus variants and support the synanthropic behavior of wild pigs in systems with severe alterations in land use, like the Laguna de Términos region.

In most cases, epidemiologic surveillance of influenza is based on antibody detection in humans and domestic animals like pigs and poultry [40,41], instead of direct detection of influenza A virions. In this sense, our study complemented serological findings with molecular detection of influenza A virus, obtaining a prevalence of 19.67%. Our prevalence level was higher than that reported in a study on wild pigs in the Innamincka Regional Reserve, Australia, in 2014; the authors of that study reported a seroprevalence of 13.04% to influenza A virus [42].

The highest positivity rate in our study was found in nasal swab samples, with a prevalence of 13.11%; this demonstrates the dissemination of virus through active shedding by infected wild pigs. On the other hand, the prevalence of 4.91% and 1.63% observed in tracheal and lung samples, respectively, proved the susceptibility of cells in the respiratory epithelium, associated to the expression of sialyl-α-2,3-Gal and sialyl-α-2,6-Gal cell receptors in pigs [43].

Another interesting finding was the statistically significant relationship between age and qRT-PCR positivity, adults being the most prevalent group; this can be a result of the innate immune response that females transfer to piglets at birth; thus, younger animals could be immunologically protected from circulating viruses, as previous studies on young wild boars have shown [44].

Additionally, the link between older age and qRT-PCR positivity could be explained considering that adult hogs have a wider roaming area. Adult hogs, specially males, cover longer distances, which favors the contact with other hosts of influenza A viruses. In our study, a male, adult hog that was positive to qRT-PCR was also positive to the HI assay for the H5N2 subtype, with a titer of 80 AHU; this suggests previous contacts with other transmitting species. A roaming area of 0.3 a 7.2 ± 1.8 km^2^ for females and of 3.1 to 48.3 ± 4.3 km^2^ for males has been reported [9,10,24,25,26]. A negative effect of a wider distribution range of wild pigs is the possible interaction with wild, domestic, and human hosts due to the transformation of ecosystems for agricultural or residential use [9,45].

In our study, sequencing and phylogenetic analysis demonstrated that the virus founded in wild pigs, despite having an avian origin, is more related to sequences reported in domestic pigs, so it is possible that transmission has occurred from domestic pig to wild pig, as shown by the sequence of the virus Feral swine/Campeche/DMZC-DEFSAL-UIFMVZ19-12 (H5N2), which was transmitted from one host to another one. In 2008, two avian H5N2 influenza virus variants were isolated from domesticated pigs. Sequencing and phylogenetic analysis of the expressed proteins revealed that the viral isolate Sw/Korea/C12/08 (H5N2) was a completely avian virus, circulating in wild birds [46]; in contrast, the avian H5N2 virus found in our study in wild pigs is a low-pathogenicity virus variant circulating in poultry, first reported in Mexico in 1994 [47].

Along with avian viruses, the clade where the virus isolated in our study is located also includes two H5N2 viruses of avian origin (broiler chicken) but isolated from domesticated pigs, GtoDMZC02 (H1N1) and EdoMexDMZC03 (H5N2), isolated and characterized in 2014 and 2015, respectively [22]. Saavedra et al. (2018) pointed out the possible interaction occurring in mixed farming systems, where poultry and pigs are in close contact, as a plausible explanation of the interspecies leap of the H5N2 avian virus from birds to swine.

Globally, one of the main risk factors for the transmission of influenza virus variants is the existence of animal production facilities where different species share a common space, favoring interspecies interaction and the transmission of influenza A viruses [48]. Additionally, the dogs (*Canis lupus familiaris*) commonly found in rural communities in Mexico can play a role in the interspecies propagation of H5N2 influenza viruses. In 2013, a novel avian/swine H5N2 reassortant influenza virus was reported to infect dogs. Initially, the virus propagated among canids, which showed respiratory signs and seroconversion, as well as virus shedding, as detected in nasal swabs; then, virus-related respiratory clinical signs were observed in cats (*Felis catus*) and chickens (*G. gallus*) [45,49]. Thus, the participation of dogs in the transmission of the H5N2 virus cannot be ruled out, since the rate of contact between dogs, wild pigs, and various types of birds is very high.

The emergence of viral exposition and infection has been associated to radical alterations in ecosystems, like deforestation and changes in the use of land for agricultural activities [50]. The current levels of human-ecosystem interaction, boosted by an increasing environmental invasion and changes in the use of land (natural resource exploitation, agricultural and livestock activities), as well as environmental effects like climatic change, have caused major habitat alterations, changes in species blending and contact rates that promote the occurrence of zoonotic diseases [33]. The link between forest fragmentation in Africa and the outbreaks of Ebola virus diseases (EVD) was measured in 2017. That work showed that most EVD outbreaks in humans occurred in forest fragmentation foci, where interactions between humans and EVD reservoirs were more frequent. As a result of higher contact rates, the frequency of EVD outbreaks in humans increased [51]. On this respect, our study, whilst being a first approximation that will be complemented with a higher number of samples and a specific experiment design, demonstrates the effect of landscape anthropization on the exposure of wild pigs to the influenza A virus in areas with a higher anthropogenic transformation and fragmentation, like agricultural and livestock land.

Currently, the Laguna de Términos region is highly exposed to deforestation due to the pressure from human population growth, which in turn results in a higher demand for food, with agricultural and livestock activities expanding in tropical regions [52,53]. Due to this anthropization, wild pigs play a key role in the eco-epidemiology of influenza A virus strains. Since they are susceptible hosts and they could have potential capacity of generating novel virus variants.

## 5. Conclusions

This study suggests that wild pigs in the Laguna de Términos region, Campeche, are participating in the exposure, transmission, maintenance, and possible diversification of influenza A virus subtypes. High antibody titers against human H1N1 virus subtype were detected, up to 2560 HIU, as well as positive titers for the swine H1N1 and avian H5N2 subtypes, which directly reflect the rate of contact of wild pigs with domestic and human hosts and evince the synanthropic behavior of wild pigs. The sequencing of the avian H5N2 virus subtype further supports the contact of wild pigs with domestic hosts like poultry and their potential role in diversifying influenza A viruses. Adult animals seem to have a key role in the ecology of influenza virus in this system. The ancestral behavior of wild pigs allows them to cover great distances, thus increasing the likelihood of interaction with wild, domestic, and human hosts, which may act as infection sources. Landscape anthropization seems to influence the seroprevalence of influenza A virus subtypes. It is likely that, as landscape anthropization proceeds, the possibility of finding wild pigs that were exposed to influenza A virus variants of wild, domestic, and human origin increases as well.

## Figures and Tables

**Figure 1 viruses-12-00528-f001:**
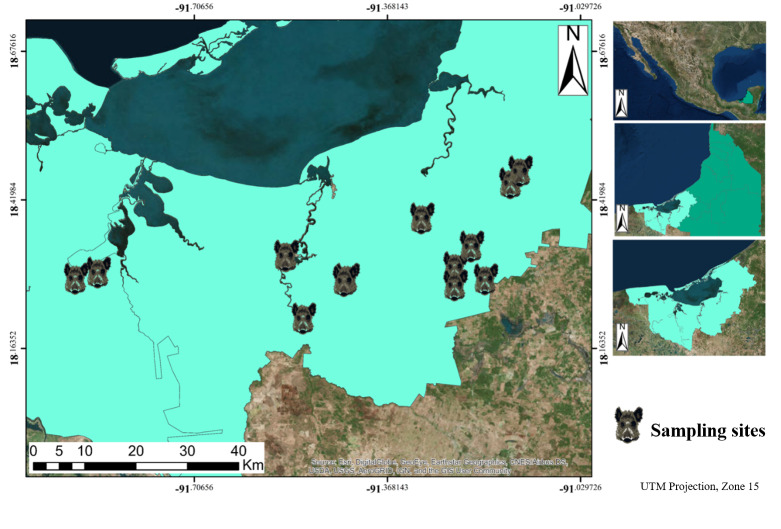
Area under study and sampling sites. Hog heads show sampling sites around the lagoon.

**Figure 2 viruses-12-00528-f002:**
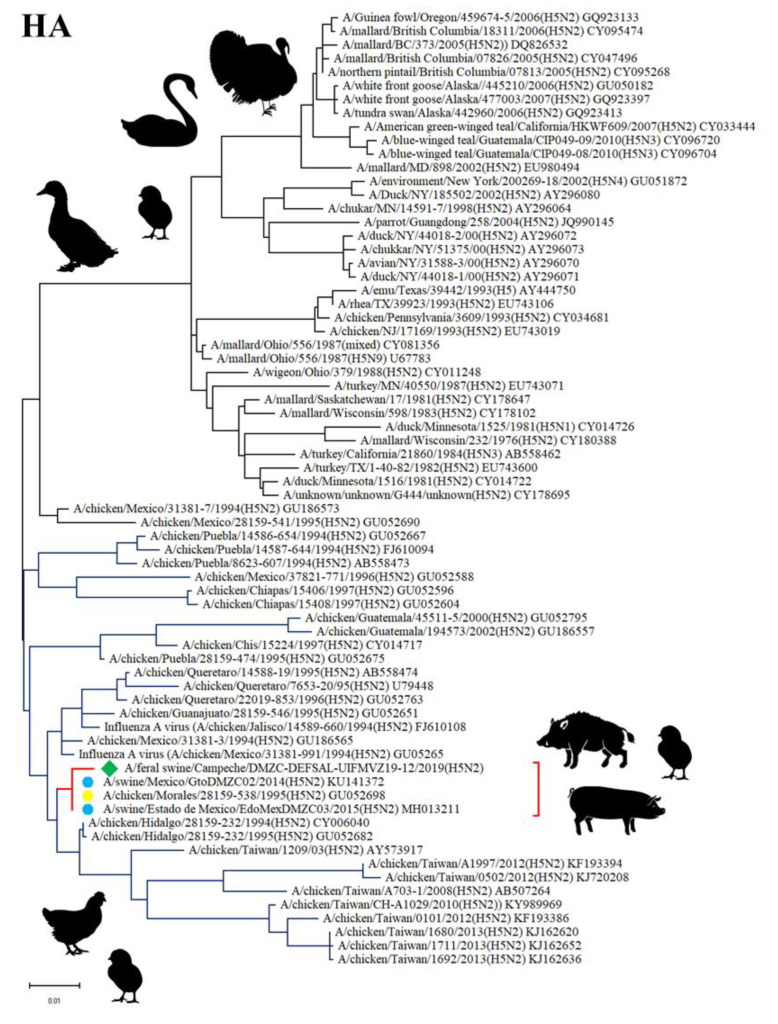
Phylogenetic tree of the *HA* gene of H5N2 influenza virus. The tree was drawn to scale, with branch length proportional to the number of substitutions per site. The H5N2 isolate herein reported is marked as ◆, while swine H5N2 subtypes are marked as ●, and the avian H5N2 virus is marked as ●.

**Figure 3 viruses-12-00528-f003:**
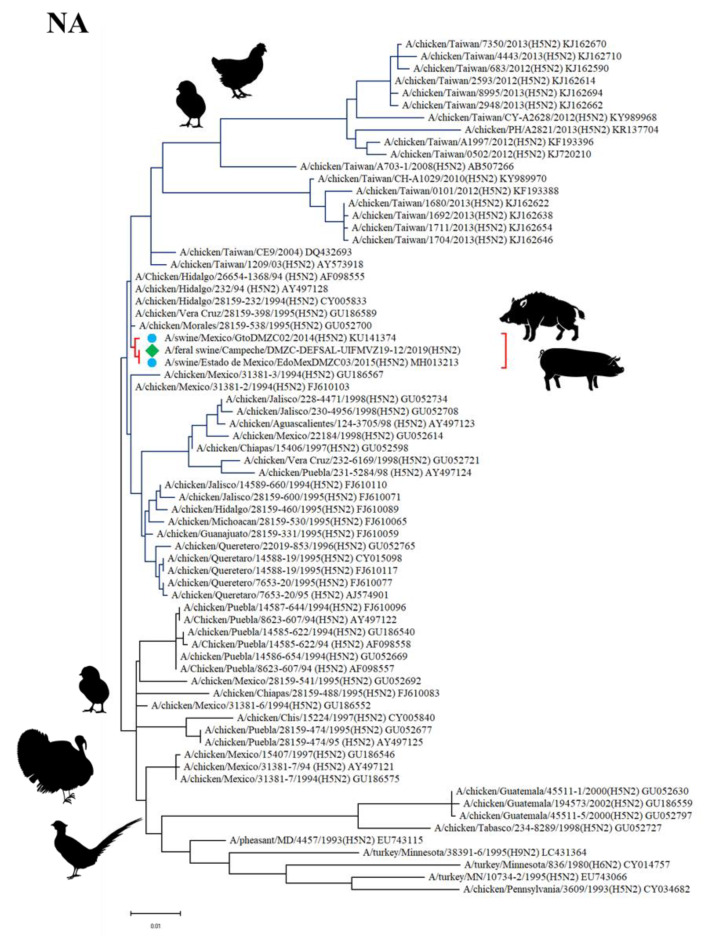
Phylogenetic tree of the *NA* gene of H5N2 influenza virus. The tree was drawn to scale, with branch length proportional to the number of substitutions per site. The H5N2 isolate herein reported is marked as ◆, while swine H5N2 subtypes are marked as ●.

**Figure 4 viruses-12-00528-f004:**
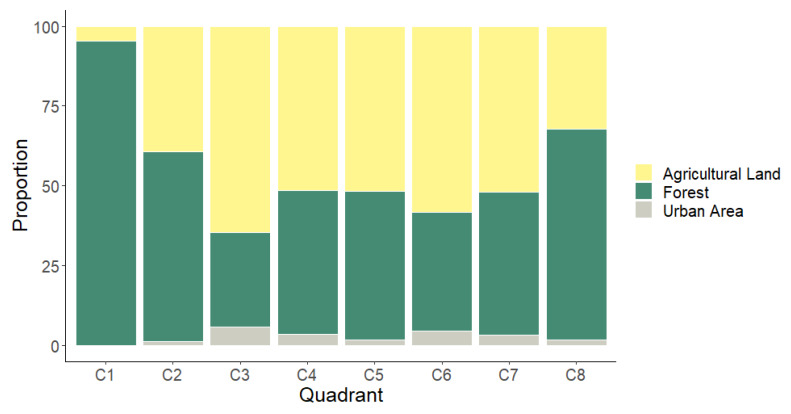
Proportion of vegetation cover by quadrant.

**Figure 5 viruses-12-00528-f005:**
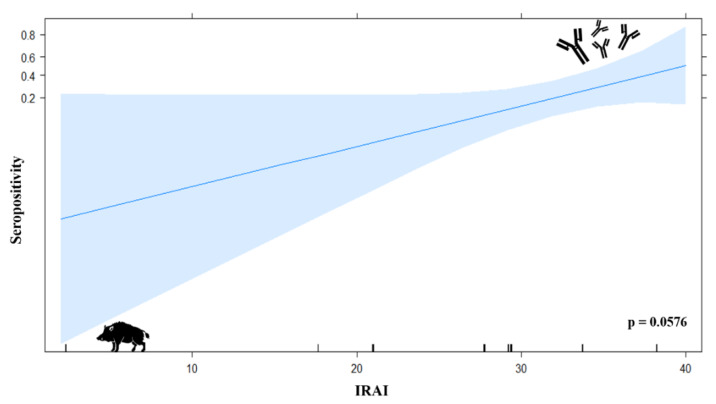
Logistic regression model. Integrated relative anthropization index (IRAI) values are shown on the *x*-axis, and seropositivity are shown on the *y*-axis. The probability that an individual is positive increases as IRAI increases. Confidence interval is shown in pale blue. Initially wide, it becomes narrower as IRAI increases. The behavior of the confidence interval is related to the sample size.

**Table 1 viruses-12-00528-t001:** Count of wild pig serum samples positive to the hemagglutination inhibition assay for at least one influenza A virus subtype. Seropositivity, frequency, antibody titer, sex, and age are shown for the total population sampled and for the five influenza subtypes used in the assay.

Variable	*n*	H1N1hu *	HI1N1sw *	H1N2sw *	H3N2sw *	H5N2av *
Total	61					
Seropositivity (Frequency % [CI: 5.85–23.65])	9 (14.75)	7 (11.47)	5 (8.19)	0 (0)	0 (0)	1 (1.63)
Ab titer (mean) Range of observations		595 80–2560	144 80–320	0 10	0 10	80 80
Sex						
Female	4	4	2	0	0	0
Male	5	3	3	0	0	1
Age						
Piglet	4	4	3	0	0	0
Juvenile	2	2	0	0	0	0
Adult	3	1	2	0	0	1

H1N1hu * = Human subtype, H1N1sw * = swine subtype, H1N2sw * = swine subtype, H3N2sw * = swine subtype, H5N2av * = avian subtype. The cutoff point to regard a serum sample as positive was 80 HIU.

**Table 2 viruses-12-00528-t002:** Count of wild pig samples positive to qRT-PCR assay. Positivity, frequency, Ct value for qRT-PCR, sex, and age are shown for the total population sampled and for nasal swab, trachea, and lung samples.

Variable	*n*	Nasal Swab	Trachea	Lung
Total	61			
Positive (Frequency % [CI: 9.70–29.65])	12 (19.67)	8 (13.11)	3 (4.91)	1 (1.63)
Ct value (mean) Range of observations		32.94 28.43–34.36	32.43 31.20–34.39	34.19 34.19
Sex				
Female	6	4	1	1
Male	6	4	2	0
Age				
Piglet	3	3	0	0
Juvenile	2	2	0	0
Adult	7	3	3	1

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
