# Peer review of "Eco-Epidemiological Evidence of the Transmission of Avian and Human Influenza A Viruses in Wild Pigs in Campeche, Mexico"

_viruses, 2020, doi:10.3390/v12050528_

Round 1

Reviewer 1 Report

The manuscript is well written and clear.

A few minor suggestions:

Line 111 : but instead of buy ?

Section 2.3. : maybe it could be useful to have some pictures of wild pigs representative of Categories A and B, as supplementary data.

Section 2.3 and/or 3.1.: there is no indication of the period of the study. Based on the strain name in Figure 2, it seems the study was conducted in 2019, but this information is not obvious. Information, such as the time of the year, the duration of the sampling period, etc.. should be indicated in at least one of these two section.

Section 2.10: would it be possible to provide some maps or illustrations/schematic to explain the process, the 8 quadrants for example. It would greatly help readers not familiarised with this type of analysis.

61 animals were sampled in this study. Is there an estimate of the whole wild pig population in the area studied? This would help to evaluate how representative is the sample collection.

Section 3.5.: for PB2, PB1, PA, NP, M, and NS genes analysis, each phylogenic trees should be presented as supplementary data. They are just mentioned in the text, line 309, at the moment.

Section 3.7.2.: age and qRT-PCR are dependent, could the nature of this result explained (i.e. increased qRT-PCR positive results in adults)?

Figure 5: y-axis, is it serology titre or seropositivit? The figure legend and the text do not agree. This should be clarified.

Reviewer 2 Report

Virus journal has a special Issue of "Influenza Viruses in Wildlife". Otherwise, I don’t think this article is suitable for issuing in Virus Journal since it contains a small amount of virology. The important parts of this article are related to environment. Infection of pigs with human and bird influenza viruses has been reported in other papers, like the references 22, 29 and 46. The relationship between serological results and IRAI values is not strong either. However, the influenza viruses were isolated from wild pigs fitting the requirement of this special issue.

Some comments are as follows.

Introduction

Line 46 Add s after give in ‘which give

Results

Line 234 titer higher than 80 HAU’ the serological titer is done by Hemagglutination inhibition, so the titer should be expressed as HI but not HA so chang HAU into HI (Hemagglutination inhibition).

Table 1 needs rearranged, for example, item on the second column “n” means number but not percentage, 14.75%. The same for Table 2.

Materials and Methods

Line 131 again, change higher than 1:80 HAU into higher than 1:80 HIU since it is antibody titer using hemagglutination inhibition (HI) assay. HAU means antigen but no antibody titer. Chang all antibody titer into HIU in the whole context.

Results

Lin 323 Change UA into AU in “The UA with highest IRAI

Line325: There was no Table 3 “and roads (Table 3).

Line 332 “variable showed a similar behavior.” Use another word to replace behavior

Discussion

Line 350 change 2560 AHU into 2560 HIU.

Line 375, “ documented do date” what does this mean?
